# Exercise-Induced Oxygen Desaturation during the 6-Minute Walk Test

**DOI:** 10.3390/medsci8010008

**Published:** 2020-01-31

**Authors:** Raghav Gupta, Gregg L. Ruppel, Joseph Roland D. Espiritu

**Affiliations:** 1Department of Pulmonary and Critical Care Medicine, Deaconess Health System, Evansville, IN 47747, USA; 2Indiana University School of Medicine, Evansville, IN 47710, USA; 3Division of Pulmonary, Critical Care, and Sleep Medicine, Saint Louis University School of Medicine, Saint Louis, MO 63103, USAjoseph.espiritu@health.slu.edu (J.R.D.E.)

**Keywords:** six-minute walk test, oxygen desaturation, predictors

## Abstract

The 6-minute walk test (6MWT) is not intended to document oxygen (O_2_) desaturation during exertion but is often used for this purpose. Because of this, it only has modest reproducibility in determining the need for ambulatory O_2_ therapy in patients with cardiopulmonary disease. The diagnostic and prognostic value of detecting exertional O_2_ desaturation is still unknown. The aims of this study were to estimate the prevalence of O_2_ desaturation during a 6MWT based on pulse oximetry measurements at the beginning and end of a 6MWT in a clinical population of patients with suspected cardiopulmonary disease and to determine whether the pulmonary function test (PFT) can predict exercise-induced desaturation during a 6MWT. This retrospective cohort study reviewed the results of the 6MWT and the PFT (i.e., spirometry, lung volumes, and diffusion capacity) of all patients who were evaluated for suspected cardiopulmonary disease at an academic medical center during a 5-year study period. The patients were categorized into three groups based on the change in O_2_ saturation by pulse oximetry (SpO_2_) from start to end of the 6MWT: (1) SpO_2_ decreased by ≥3%; (2) SpO_2_ unchanged (−2 ≤ Δ ≤ 0%); and (3) SpO_2_ increased by ≥1%. Demographic, anthropometric, and lung function measurements were analyzed to determine which factors predicted O_2_ desaturation during the 6MWT. Of the 319 patients who underwent the 6MWT and the PFT from November 2005 until December 2010 (mean age = 54 ± 0.78 years, 63% women, 58% Whites, body mass index = 29.63 ± 8.10 kg/m^2^), 113 (35%) had a decreased SpO_2_, 146 (46%) had no change, and 60 (19%) had an increased SpO_2_ from the start to end of test. Our bivariate analysis found age, spirometric measures, and diffusion capacity for carbon monoxide (DLCO) had statistically significant inverse associations with the SpO_2_ change category (*p* < 0.001). Both a 3% and 4% drop in SpO_2_ during the 6MWT were statistically significantly associated with an older age, a higher prevalence of obstruction, and reduced forced vital capacity (FVC), forced expiratory volume in one second (FEV_1_), FEV_1_/FVC, DLCO and 6-minute walk distance (6MWD). Multivariable logistic regression analyses revealed that only DLCO was a significant independent predictor of the change in SpO_2_ and a ≥ 4% O_2_ desaturation during a 6MWT. Receiver operating curve analysis indicates DLCO cut-off of 45% is 82% sensitive and 40% specific in identifying ≥4% O_2_ desaturators, with an area under the curve of 0.788 ± 0.039 (*p* < 0.001). The prevalence of a ≥ 3% oxygen desaturation via pulse oximetry during a 6MWT in our clinical population of patients with suspected cardiopulmonary disease was 35%. Although age, spirometric lung volumes, and DLCO had statistically significant unadjusted inverse associations with the change in SpO_2_ during a 6MWT, the DLCO is the only significant independent predictor of both the magnitude of the change in SpO_2_ and the occurrence of O_2_ desaturation of at least 4%, respectively, during the test. Clinical Implications: A DLCO cut-off of 45% may be useful in identifying patients at risk for exertional hypoxemia during a 6MWT.

## 1. Introduction

Functional exercise capacity is an important measure for evaluating and monitoring patients with cardiopulmonary disease. Various modalities have been used to measure exercise capacity including a 6-minute walk test (6MWT), stair climbing, a shuttle-walk test, and a cardiac pulmonary exercise testing (CPET) [1]. The 6MWT has been broadly used in clinical settings for its patient tolerability, ease of implementation without need for special equipment and good correlation with patient outcomes [2,3]. It provides information regarding functional capacity, response to therapy and prognosis across a range of chronic cardiopulmonary conditions [4]. The 6MWT yields three main measurements: 6-minute walk distance (6MWD), exercise-induced oxygen desaturation (EID), defined as an oxygen saturation by pulse oximetry (SpO_2_) to ≤88%, and self-perceived dyspnea as assessed by the Borg scale [5]. It is well known that the 6MWD is a predictor of mortality, primarily in patients with cardiopulmonary disease [6]. In addition, patients who developed oxygen (O_2_) desaturation during the 6MWT had a higher mortality rate than patients who did not (67 vs. 38%, *p* < 0.001) [7].

The 6MWD has been known to correlate well with pulmonary function test (PFT) variables including forced expiratory volume in one second (FEV_1_), forced vital capacity (FVC) and diffusion capacity for carbon monoxide (DLCO) [8]. Chen et al. found FEV_1_ to be a predictor of 6MWD in stable chronic obstructive pulmonary disease but was not associated with either O_2_ desaturation or dyspnea measured on the BORG scale [9]. The FVC and DLCO were found to be strongly correlated with 6MWD in idiopathic pulmonary fibrosis (IPF) patients [10]. A DLCO <50% of predicted and an FVC <80% of predicted were the best parameters to identify abnormal 6MWD in patients with scleroderma-associated interstitial lung disease (ILD) [11]. Although PFT variables appear to correlate well with the 6MWD, less is known about the prediction of O_2_ desaturation [12].

Poulain et al. demonstrated the reproducibility of EID during the 6MWT patients with lung disease when compared to CPET [13]. Guyatt and colleagues also found that measuring O_2_ saturation by pulse oximetry (SpO_2_) during a 6MWT could establish a baseline value for ascertaining improvements in SpO_2_ during repeated testing [14]. The American Thoracic Society (ATS) 2002 6MWT guidelines made a clear statement of not measuring O_2_ continuously throughout the test due to motion artifact [4,15]. After 2002, multiple studies have shown the reliability of measuring SpO_2_ during a 6MWT [16,17]. An updated statement from European Respiratory Society (ERS) and ATS now recommends using 6MWT as a standard O_2_ desaturation test with the use of a pulse oximeter for continuous measurement of the SpO_2_ and heart rate (HR) [8,18].

Previous studies have observed O_2_ desaturation to occur commonly during the 6MWT [19,20,21]. Paciocco et reported that 18 of 34 (53%) of pulmonary hypertensive patients experienced a ≥ 10% O_2_ desaturation during the 6MWT and an almost 3-fold increased mortality over a 5 year follow up. ^19^ Jenkins et al. reported a 47% prevalence of significant O_2_ desaturation defined as a ≥4% fall in SpO_2_ to <90% in a large cohort of patients with chronic lung disease, with pre-exercise SpO_2_ and forced expiratory volume in 1 s (FEV_1_) as significant predictors of desaturation [20]. An analysis of the 6MWT in cystic fibrosis patients with mild-to-moderate lung disease by Chetta et al. identified forced expiratory volume in 1 s (FEV_1_) as the only independent predictor of SpO_2_ during a 6MWT [21]. Knowledge of which PFT measurements predict O_2_ desaturation during exercise can assist the clinician in identifying patients who may require O_2_ supplementation during exercise, in order to relieve exertional dyspnea and enhance exercise capacity. Thus, the aims of this study were to estimate the prevalence of O_2_ desaturation during a 6MWT based on SpO_2_ measurements from the beginning and end of the 6MWT in a clinical population of patients with suspected cardiopulmonary disease and to determine whether the PFT can predict EID during a 6MWT.

## 2. Materials and Methods

### 2.1. Patient Selection

This retrospective cohort study evaluated all adult patients who underwent the PFT and the 6MWT for various clinical indications (e.g., cardiopulmonary disorders) from November 2005 until December 2010. The only exclusion criterion was the absence of either PFT or 6MWT results. The patients were initially categorized into 3 groups based on the magnitude of the change in SpO_2_ from the start to the end of the 6MWT (Δ = SpO_2_ at end of test–SpO_2_ at start of test): (1) SpO_2_ increased (Δ ≥ +1%); (2) SpO_2_ unchanged (−2 ≤ Δ ≤ 0%); and (3) SpO_2_ decreased (Δ ≤ −3%). Since the existing literature was unclear as to the range of normal change (Δ) in SpO_2_ during a 6MWT, we selected an arbitrary cut-off of ≤−3% to define the “SpO_2_ decreased” group which experienced clinically significant decrements in SpO_2_ during the test. The rationale for the −3% cut-off is that most pulse oximeter manufacturers claim an accuracy of 2%, which is the standard deviation of the differences between the SpO_2_ by pulse oximetry and the SaO_2_ by CO-oximetry [22]. Assuming that moderate exercise during a 6MWT should not result in any significant change in oxygenation in normal individuals, patients whose SpO_2_ changes were minimal (−2 to 0%) where categorized under the “SpO_2_ unchanged” group. We assigned an arbitrary cut-off of ≥+1% to categorize those patients whose SpO_2_ increased during moderate exercise with the purpose of identifying patients whose ventilation-perfusion matching might improve with the increase in tidal volume, heart rate, and stroke volume associate with moderate exercise. We also binarily categorized the patients as “oxygen desaturators” vs. “non-desaturators” based on whether the patient had an SpO_2_ drop of at least 4% during the 6MWT.

### 2.2. Measurements

#### Demographic and Anthrompometric Variables

Demographic data (i.e., age, sex, and race) and anthropometric measurements (i.e., height, weight, body mass index or BMI) at the time of the PFT were retrospectively collected from the PFT technical report and/or 6MWT.

### 2.3. Pulmonary Function Test Protocol

The PFT was conducted following the recommendations of the ATS and ERS Task Force on the standardization of spirometry and measurement of lung volumes and their technical statement on the measurement of the single-breath carbon monoxide uptake in the lung, respectively. Pre-bronchodilator forced vital capacity (FVC), forced expiratory volume in 1 s (FEV_1_), FEV_1_/FVC ratio, total lung capacity (TLC), residual volume (RV), and diffusion capacity for carbon monoxide (DLCO) were collected in our data file. The DLCO value was adjusted for hemoglobin by applying Cotes’ equation: adjusted DLCO = [DLCO × (10.2 + hemoglobin)/(1.7 × hemoglobin)].
We categorized patient’s PFT results as “obstruction” if the FEV_1_/FVC ratio is less than 0.70; “restriction” if the total lung capacity is less 80% of predicted; and “combined obstruction and restriction” if both are present.

### 2.4. The 6-Minute Walk Test Protocol 

The 6MWT was performed in accordance with ATS guidelines published in March 2002. The 6-minute walk distance (6MWD) and the SpO_2_, Borg dyspnea score, Fatigue score, and pulse rate at the start and end of the test were collected in the data file. Pulse oximetry SpO_2_ was measured via a finger probe using a Masimo Rad 5 with Adult Sensor (Masimo, Irvine, CA, USA) while standing at the beginning of the 6MWT and then again while standing at the end of the test after 6-min of walking. For patients who had never been prescribed O_2_ supplementation, pulse oximetry was conducted on room air. For patients who had previously required O_2_ supplementation, pulse oximetry was performed on their prescribed O_2_ flow rate via nasal cannula or fraction of inspired O_2_ (F_i_O_2_) via mask.

### 2.5. Statistical Analysis

We calculated the prevalence of EID, defined as a drop of at least 3 or 4% in SpO_2_ from the start to the end of the 6MWT. Analysis of variance (ANOVA) and *χ*^2^ tests were employed to determine whether demographic (i.e., age, sex, and race) and anthropometric (i.e., height, weight, and BMI) characteristics, PFT variables (i.e., pre-bronchodilator FVC, FEV_1_, FEV_1_/FVC; TLC, RV, DLCO, and airway resistance), lung function abnormalities (i.e., “obstruction”, “restriction”, or “combined obstruction and restriction”), and 6MWD were associated with the (1) category of SpO_2_ change (i.e., increased, unchanged, or decreased) and (2) a ≥ 4% O_2_ desaturation (≥4% Desaturators vs. Nondesaturators), respectively. Repeated Measures ANOVA was also conducted to determine whether the change in 6MWT measures (i.e., SpO_2_, Borg Dyspnea and Fatigue Scale scores, and pulse rate) from the start to the end of the test were statistically different among categories of SpO_2_ change (i.e., increased, unchanged or decreased) and between ≥4% Desaturators and Nondesaturators, respectively. Multivariable linear regression analysis was conducted to determine which of the significant independent demographic, anthropometric, and PFT variables were independently associated with the magnitude of the SpO_2_ change (Δ). We also fitted multivariable binary logistic regression models to determine whether the significant independent demographic, anthropometric, and PFT variables predicted the occurrence of a ≥ 3 and 4% desaturation, respectively, (in order to distinguish desaturators vs. non-desaturators) during a 6MWT. Statistical analyses were performed using SPSS software (IBM SPSS Statistics for Windows, Version 22.0. Armonk, NY, USA: IBM Corp. Released 2013).

## 3. Results

### 3.1. Demographic, and BMI Characteristics of Patients Based on Category of SpO_2_ Change during the 6MWT

Of the 319 patients who underwent the PFT and the 6MWT for suspected cardiopulmonary disease at our academic medical center from November 2005 until December 2010, there were 206 women (63%) and 122 men (37%). The racial distribution of our study participants reflected our Midwestern urban location of our academic center, with 184 (57%) Whites, 129 (41%) African-Americans, three (1%) other races, and three (1%) unknown race. The mean ± standard deviation (SD) age was 54 ± 0.78 years and the mean BMI was 29.63 ± 8.10 kg/m^2^. Table 1 compares the demographic, clinical, and pulmonary function findings based on the category of SpO_2_ Δ during a 6MWT. Of the 319 study patients, 113 (35%) experienced an SpO_2_ decrease of at least 3% from the start to the end of the 6MWT. In contrast, 146 (46%) had a minimal-to-no change (SpO_2_ Δ = 0 to −2%), while 60 (19%) achieved an increase of at least +1%. Age had a statistically significant inverse relationship with SpO_2_ Δ category, with the groups whose SpO_2_ decreased (57.68 ± 13.91 years) or remain unchanged (54.41 ± 13.35 years) being older than those whose SpO_2_ increased (48.30 ± 13.96 years, *p* ≤ 0.01 on post-hoc comparisons). There were no statistically significant differences in the sex and race distribution and BMI among the three SpO_2_ change categories (*p* > 0.05).

### 3.2. Pulmonary Function of Patients Based on Category of SpO_2_ Change during the 6MWT

Spirometric lung volumes were also significantly associated with the SpO_2_ Δ category, and the group whose SpO_2_ decreased by at least 3% had lower FVC (*p* < 0.001), FEV_1_ (*p* < 0.001), and FEV_1_/FVC ratios (*p* < 0.014) compared to the groups whose SpO_2_ were unchanged or increased, respectively. On the other hand, there were no statically significant differences in lung volumes (i.e., TLC and RV) or airway resistance by body plethysmography (*p* > 0.05) among the three groups. The group whose SpO_2_ decreased by at least 3% also had a statistically significantly lower DLCO than the groups whose SpO_2_ was unchanged (*p* < 0.001) or increased (*p* = 0.001), respectively. The prevalence of obstruction based on an FEV_1_/FVC ratio <0.70 was also significantly higher in the SpO_2_ decreased group (46%) than the SpO_2_ unchanged (26%) and increased (22%) groups, respectively (*p* < 0.001). There was no significant difference in the prevalence of restriction (TLC < 80%) or combined obstruction and restriction among the categories of SpO_2_ change (*p* > 0.05).

### 3.3. The 6-Minute Walk Test Results of Patients Based on Category of SpO_2_ Change during the 6MWT

Table 2 lists the results of the 6MWT for the three groups of patients categorized based on the change in SpO_2_ (increased, unchanged, or decreased) from the start to the end of the test. ANOVA revealed statistically significant differences in the SpO_2_, both at baseline and at the end of the test, among the three categories of SpO_2_ change. Although the baseline SpO_2_ values were statistically different among the three categories of SpO_2_ change, these differences were not clinically meaningful (increased group: 96.57 ± 2.04%, unchanged group: 97.99 ± 1.89%, and decreased group: 97.20 ± 2.43%). On the other hand, the SpO_2_ at the end of the test was both statistically and clinically significantly lower (*p* < 0.001) in the group that had at least a −3% decrease in SpO_2_ (90.42 ± 5.11%) compared to the increased group (98.45 ± 1.63%) and the unchanged group (97.10 ± 2.10%), respectively. Repeated Measures ANOVA found a statistically significant association between the change in SpO_2_ from the start to the end of the 6MWT and the three categories of SpO_2_ change. The group that had an increase in O_2_ saturation by at least 1% from start to end of the 6MWT (SpO_2_ Δ ≥1%) had an average +1.88% rise; the unchanged group, whose oxygen saturation dropped anywhere from 0 to −2% from baseline (−2 ≤ Δ ≤ 0%), had a mean −0.89% drop; and the group that had at least a −3% drop from baseline (Δ ≤ −3%) had a mean −6.78% drop in SpO_2_ from the start to the end of the test (*p* < 0.001). In addition, there was a statistically significant inverse association between the 6MWD and the category of SpO_2_ change (*p* = 0.033). On post-hoc comparison, the 6MWD tended to be shorter in the group whose SpO_2_ decreased by at least 3% (353.98 ± 138.27 m) compared to the groups whose SpO_2_ were either unchanged (392.99 ± 119.17 m, *p* =0.05) or increased (397.88 ± 147.42 m, *p* = 0.097), respectively. There were no statistically significant differences in Borg dyspnea scores, Fatigue scores, or pulse rates either at the start or end of the test among the three categories of SpO_2_ change.

### 3.4. Demographic, and BMI Characteristics of ≥4% Oxygen Desaturators vs. Nondesaturators during the 6MWT

After categorizing participants based on the presence of the occurrence of a ≥ 4% drop in SpO_2_, 95 of 319 subjects (30%) were classified as ≥4% Desaturators while 224 (70%) as Nondesaturators (Table 3). Age, spirometric lung volumes, and DLCO were statistically significantly associated with a drop of ≥4% in SpO_2_. The ≥4% Desaturators were statistically significantly older (57.96 ± 13.20 vs. 52.92 ± 14.11 years), were more likely to have obstruction (FEV_1_ < 70%), and had lower FVC, FEV_1_, FEV_1_/FVC ratio, and DLCO (*p* < 0.05) Sex, race, the prevalence of restriction or combined obstruction-restriction, RV, TLC, and airway resistance were not significantly different between ≥4% Desaturators and Nondesaturators.

### 3.5. The 6-Minute Walk Test Results of ≥4% Oxygen Desaturators vs. Nondesaturators during the 6MWT

The baseline SpO_2_, Borg Dyspnea Scale score, Fatigue score and pulse rate were similar between ≥4% Desaturators and Nondesaturators (Table 4). At the end of the 6MWT, the ≥4% Desaturators were statistically significantly more hypoxemic than the Nondesaturators (SpO_2_: 89.67 ± 5.15 vs. 97.24 ± 2.26%, *p* < 0.001). The ≥4% Desaturators also had a statistically shorter 6MWD (352.67 ± 138.19 vs. 391.69 ± 128.83 m, *p* = 0.017). There were no statistically significant differences in the changes in Borg Dyspnea and Fatigue Scale scores and pulse rate from the start to the end of the 6MWT between the ≥4% Desaturators and Nondesaturators.

### 3.6. Regression Models to Predict Changes in SpO_2_ and a ≥4% Oxygen Desaturation during the 6MWT

A multivariable linear regression model was fit to determine whether the magnitude of the change in SpO_2_ (as a continuous variable) during the 6MWT was associated with the statistically significant independent variables (i.e., age, FVC, FEV_1_, and DLCO) in the unadjusted analysis (Table 5). Although the FEV1/FVC ratio and the presence of obstruction based on the same ratio were found to be statistically significant in the bivariate analysis, we excluded them in the model in order to avoid multicollinearity. This linear regression model revealed that after adjusting for age, FEV_1_, and FVC, the DLCO was the only statistically significant predictor of the change in SpO_2_ during a 6MWT (*β* = −0.306 ± 0.072, *t* = −4.271, *p* < 0.001, *R*^2^ = 0.147). Based on the *R*^2^, this linear regression model using age, FVC, FEV_1_, and DLCO as covariates accounts for 15% of the variance in the change in SpO_2_ during a 6MWT.

A binary logistic regression model was initially fitted to determine whether the age, FVC, FEV_1_, and DLCO predicted the occurrence of O_2_ desaturation of ≥3% during the 6MWT (Table 6). None of the independent variables (i.e., age, FVC, FEV1, and DLCO) was found to be a statistically significant predictor of a ≥3% drop in SpO_2_ during a 6MWT. On the other hand, a second logistic regression model identified the DLCO as the only independent predictor of a ≥ 4% drop in SpO_2_ drop during a 6MWT (*β* = −0.289 ± 0.068, Wald *χ*^2^ = 17.887, *p* < 0.001, Nagelkerke *R*^2^ = 0.276) (Table 7). Based on the *Naegelkerke R^2^*, this logistic regression model using age, FVC, FEV_1_, and DLCO as covariates accounts for 28% of the variance of the outcome, i.e., a ≥ 4% drop in SpO_2_ during the 6MWT.

### 3.7. Receiver Operating Characteristics of DLCO% for Detection of ≥4% Oxygen Desaturation during the 6MWT

A receiver operating characteristic (ROC) curve was plotted to determine the performance of the DLCO% of predicted as a diagnostic test for detecting a ≥4% O_2_ desaturation during a 6MWT (Figure 1). The analysis of the ROC curve revealed a statistically significant area under the curve of 0.788 ± 0.039 (*p* < 0.001), which means that the DLCO% is a good diagnostic test to discriminate O_2_ desaturators from non-desaturators. A diagnostic DLCO% cut-off of 45% has a sensitivity of 81.8% and a specificity of 39.5%.

## 4. Discussion

This retrospective study focused on EID during a 6MWT presents two main findings: (1) the prevalence of a ≥3% oxygen desaturation via pulse oximetry during a 6MWT in a clinical population of patients with suspected cardiopulmonary disease is 35%; and (2) although age, spirometry lung volumes, and DLCO have a statistically significant unadjusted inverse association with the change in SpO_2_ during a 6MWT, the DLCO is the only significant independent predictor of both the magnitude of the change in SpO_2_ and the occurrence of O_2_ desaturation of at least 4%, respectively, during the test.

Our study found that approximately one out of three clinic patients who underwent 6MWT for suspected cardiopulmonary disease at our academic medical center developed EID of at least 3% during the test. Currently, based on our review of the literature, there are no population-based studies to define the reference ranges based on age, sex, anthropometric characteristics (height, weight, BMI) for changes in SpO_2_ during a 6MWT. We believe that our estimated prevalence may only apply to patients with moderate-to-severe pulmonary disease based on the overall mean FEV_1_% predicted (68.56 ± 22.52%) of the patients included in our analysis. Our study’s estimated 35% prevalence of ≥3% O_2_ desaturation corroborates the common occurrence of EID during a 6MWT in other studies: 53% prevalence of ≥10% desaturation in pulmonary hypertensive patients in the Paciocco study [19] and 47% prevalence of a 4% desaturation in patients with chronic lung disease in the Jenkins study [20]. The lower prevalence of O_2_ desaturation in our cohort with various cardiopulmonary disorders may be due to the inclusion of patients with less severe pulmonary function derangement.

Our unadjusted statistical analysis found a statistically significant inverse relationship between age, spirometry lung function (FVC, FEV_1_, and FEV_1_/FVC ratio), and DLCO and the category of SpO_2_ change (decreased, unchanged, increased). In other words, older age and worsening spirometry lung function and diffusion capacity are significantly associated with a ≥ 3% drop in O_2_ desaturation during a 6MWT. Our study findings corroborate those of Wilsher et al.’s, which found a strong correlation between O_2_ desaturation and both the FEV_1_ (*r* = 0.55, *p* = 0.01) and the FVC (*r* = 0.59, *p* = 0.01) in 30 patients with scleroderma lung disease who underwent 6MWT [17].

On the other hand, our multivariable linear and logistic regression analyses revealed that, after adjusting for age and spirometry lung volumes, the DLCO was the only significant independent predictor of both the magnitude of the change in SpO_2_ and the occurrence of ≥4% desaturation, respectively, during the test. In fact, our adjusted logistic regression model incorporating DLCO as predictor and age, FEV_1_, and FVC as covariates accounted for up to 28% of the variance in the outcome, i.e., occurrence of ≥4% desaturation during a 6MWT. Our ROC curve analysis also revealed that the DLCO% of predicted was a statistically significant test variable to discriminate ≥4% O_2_ Desaturators from Nondesaturators. Based on the ROC curve, we propose a diagnostic DLCO cut-off of 45% of predicted, which corresponds to a sensitivity of 81.8% and a specificity of 39.5%.

Our study finding on the strong inverse association between DLCO and the O_2_ desaturation was also corroborated by a number of published studies. Chetta et al., who reported that O_2_ desaturation during a 6MWT was predicted by DLCO, particularly with a cut-off 57% of predicted [23]. Hadeli et al. investigated the risk of O_2_ desaturation during a submaximal step exercise test in 8000 patients of various respiratory diseases. Not only did they confirm that that the risk of O_2_ desaturation was very high in patients with a low DLCO, but that a DLCO diagnostic cut-off point of <62% predicted resulted in a 75% sensitivity and specificity for EID [24]. In a retrospective study of 97 patients, the DLCO had a slightly better ability to predict O_2_ desaturation than the DLCO/VA, with a cut-off of normal being 55% predicted [25].

The early detection of EID is important in the management of patients with suspected cardiopulmonary disease since O_2_ supplementation has been demonstrated to ameliorate dyspnea and fatigue as well as enhance exercise capacity in patients with exercise-induced hypoxemia even in the absence of resting hypoxemia [26]. The early detection of a reduced DLCO on a PFT can prompt the physician to order a 6MWT to diagnose EID and subsequently prescribe ambulatory O_2_ therapy for relief of symptoms and enhancement of exercise capacity. Randomized, controlled trials with longitudinal follow up are still necessary to determine whether ambulatory O_2_ therapy for exercise-induced hypoxemia, in the absence of resting hypoxemia, is beneficial in improving survival and other long-term health outcomes in various cardiopulmonary diseases.

The strengths of our study include the large sample size, the standardized PFT and 6MWT protocols, and the statistical adjustments employed to identify significant independent predictor/s of O_2_ desaturation during a 6MWT. On the other hand, the limitations of our study are the following: (1) The retrospective study design prevented us from ascertaining the specific cardiopulmonary disease diagnoses for each of the participants, given the limited clinical information available in the PFT laboratory records for our review. Although we were able to identify patients with obstruction, restriction, or combined abnormality based on the PFT, we were unable to compare the propensity for EID during the 6MWT among specific disease etiologies. (2) Selection bias is likely present in our study since we retrospectively selected a convenience sample of clinical patients who underwent 6MWT for suspected cardiopulmonary disease. In order to limit this selection bias, we included all patients who underwent 6MWT within the designated study period and adjusted our final statistical model based on significant confounders (age and spirometric lung function). (3) Since O_2_ desaturation was only measured at the start and at end of the 6MWT, some patients who developed hypoxemia in the middle of the test have been misclassified as “Nondesaturators”. Measuring SpO_2_ continuously throughout the 6MWT was prohibited in ATS 2002 guidelines which became a formal recommendation in the update of ERS/ATS 2014 measurement properties of field walking tests in chronic respiratory disease [4,18]. Fiore et al. showed that the SpO_2_ at the end of the test and the nadir SpO_2_ were generally similar during the 6MWT in 86 subjects with chronic lung disease except in those subjects who rested during the test. Fiore’s group thus concluded that consideration should be given to the constant monitoring of SpO_2_ during the 6MWT [27]. Our 6MWT protocol did not include continuous SpO_2_ monitoring during the 6MWT, which might have resulted in failing to detect abnormally low nadir SpO_2_ in some patients, leading to the underestimation of the prevalence of EID in our study. A prospective study of both community based and clinical populations (with ascertainment of specific cardiopulmonary disease etiology) using continuous SpO_2_ monitoring during a 6MWT will help clarify both the normative ranges and pathologic values of SpO_2_ changes for this test.

In conclusion, the prevalence of a ≥3% oxygen desaturation via pulse oximetry during a 6MWT in our clinical population of patients with suspected cardiopulmonary disease is 35%. Although age, spirometry lung volumes, and DLCO all had statistically significant unadjusted inverse associations with the change in SpO_2_ during a 6MWT, the DLCO is the only significant independent predictor of both the magnitude of the change in SpO_2_ and the occurrence of O_2_ desaturation of at least 4%, respectively, during the test.

## Figures and Tables

**Figure 1 medsci-08-00008-f001:**
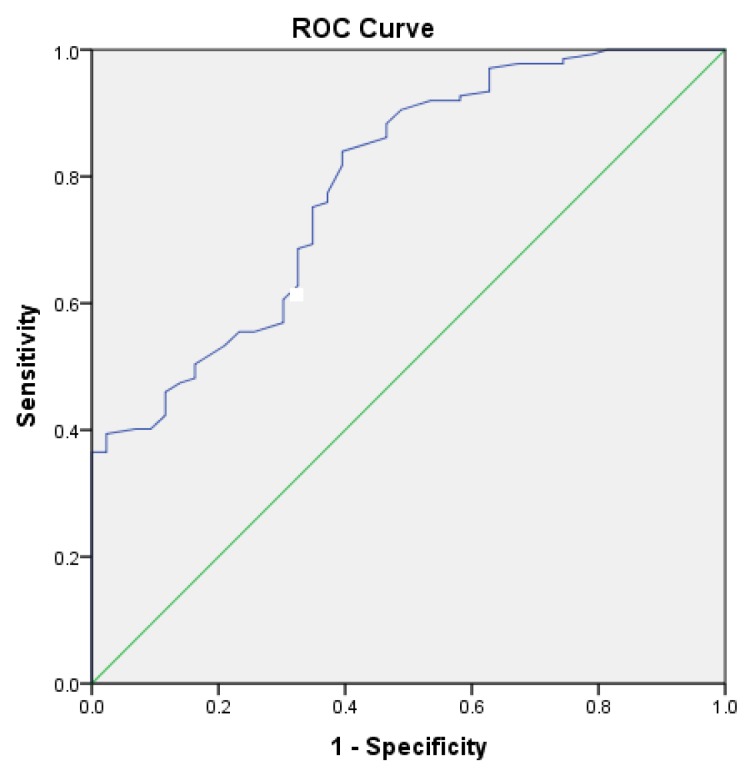
A receiver operating characteristic curve illustrating the performance of the diffusion capacity for carbon monoxide (DLCO%) of predicted as a diagnostic test for identifying patients who develop ≥4% oxygen desaturation during a 6-minute walk test. The area under the curve was 0.788 ± 0.039 (*p* < 0.001).

**Table 1 medsci-08-00008-t001:** Demographic, clinical, and pulmonary function variables based on the change in SpO_2_ category during the 6-minute walk test.

	Change in SpO_2_ (Δ = SpO_2_ at End of Study–SpO_2_ at Start of Study)			
Variable	Increased (Δ ≥ +1%) *N* = 60 (19%)	Unchanged (−2 ≤ Δ ≤ 0%) *N* = 146 (46%)	Decreased (Δ ≤ −3%) *N* = 113 (35%)	*p*-Value
Age, years	48.30 ± 13.96	54.41 ± 13.35	57.68 ± 13.91	<0.001
Sex, no. (%)				0.236
Male	22 (18)	51 (42)	47 (39)	
Female	37 (19)	95 (48)	67 (33)	
Race, no. (%)				0.364
White	35 (19)	84 (45)	66 (36)	
Black	25 (19)	59 (46)	47 (36)	
Hispanic	0 (0)	1 (100)	0 (0)	
Asian	1 (100)	0 (0)	0 (0)	
Other	0 (0)	0 (0)	1 (100)	
BMI (kg/m^2^)	27.82 ± 7.68	30.06 ± 8.28	30.06 ± 8.06	0.157
Obstruction (FEV_1_/FVC < 70%), no. (%)	12 (20)	35 (24)	46 (41)	<0.001
Restriction (TLC < 80%), no. (%)	20 (33)	37 (25)	39 (35)	0.114
Combined obstruction and restriction, no. (%)	0 (0)	2 (1)	3 (3)	0.387
FVC, L	2.92 ± 1.06	2.88 ± 0.97	2.46 ± 1.02	0.003
FVC, %predicted	76.33 ± 16.71	79.85 ± 16.53	68.56 ± 23.24	<0.001
FEV_1_ L	2.16 ± 0.93	2.10 ± 0.76	1.63 ± 0.77	<0.001
FEV_1_, %predicted	70.91 ± 23.49	74.85 ± 20.23	58.12 ± 21.89	<0.001
FEV_1_/FVC ratio	73.15 ± 16.99	73.86 ± 14.56	67.31 ± 21.21	0.014
TLC, L	4.91 ± 1.41	4.92 ± 1.47	4.67 ± 1.55	0.418
TLC, %predicted	89.12 ± 18.31	91.67 ± 21.52	85.92 ± 26.71	0.174
RV, L	2.01 ± 0.94	2.08 ± 1.01	2.23 ± 1.10	0.363
RV, %predicted	114.82 ± 53.17	113.01 ± 49.08	117.41 ± 58.12	0.825
DLCO, mL/min/mmHg	14.85 ± 5.36	15.58 ± 6.15	10.59 ± 0.94	<0.001
DLCO, %predicted	64.57 ± 21.55	71.40 ± 26.28	45.07 ± 19.25	<0.001
Airway resistance, cmH_2_O/L/sec	1.93 ± 1.35	1.88 ± 1.20	2.27 ± 1.48	0.075

SpO_2_: Saturation by pulse oximetry; BMI: body mass index; FEV1: forced expiratory volume in one second; FVC: forced vital capacity; TLC: total lung capacity; RV: residual volume; DLCO: difussion capacity for carbon monoxide.

**Table 2 medsci-08-00008-t002:** The 6-minute walk test measurements based on the change in SpO_2_ category.

	Change in SpO_2_ (Δ = SpO_2_ at End of Study–SpO_2_ at Baseline)			
6MWT Variable	Increased (Δ ≥ +1) *N* = 60 (19%)	Unchanged (−2 ≤ Δ ≤ 0%) *N* = 146 (46%)	Decreased (Δ ≤ −3%) *N* = 113 (35%)	*p*-Value
SpO_2_, %				
Start of test	96.57 ± 2.04	97.99 ± 1.89	97.20 ± 2.43	<0.001 ^1^
End of test	98.45 ± 1.63	97.10 ± 2.10	90.42 ± 5.11	<0.001 ^1^
Change	+1.88	−0.89	−6.78	<0.001 ^2^
Borg Dyspnea				
Start of test	1.52 ± 1.64	1.20 ± 1.38	1.00 ± 1.34	0.076
End of test	3.51 ± 1.93	3.16 ± 2.23	3.86 ± 6.97	0.465
Change	+1.99	+1.96	+2.86	0.611
Borg Fatigue				
Start of test	1.43 ± 1.90	1.48 ± 1.78	1.28 ± 1.65	0.660
End of test	3.51 ± 3.07	3.59 ± 2.69	2.82 ± 2.31	0.054
Change	+2.08	+2.11	+1.54	0.132
Pulse rate, 1/min				
Start of test	79.13 ± 15.26	79.31 ± 15.96	80.73 ± 14.53	0.714
End of test	103.88 ± 24.07	105.66 ± 20.71	113.77 ± 78.72	0.331
Change	+24.75	+26.35	+33.04	0.312
6-minute walk distance, m	397.88 ± 147.42	392.99 ± 119.17	353.98 ± 138.27	0.033 ^1^

^1^*p*-value from comparison based on categories of the change in SpO_2_ by ANOVA. ^2^
*p*-value from comparison based on categories of the change in SpO_2_ by Repeated Measures ANOVA.

**Table 3 medsci-08-00008-t003:** Demographic, clinical, and pulmonary function variables in ≥4% oxygen Desaturators vs. Nondesaturators during the 6-minute walk Test.

Variable	Nondesaturators (Δ SpO_2_ > −3%) *N* = 224 (70%)	Desaturators (Δ SpO_2_ ≤ −4%) *N* = 95 (30%)	*p*-Value
Age, years	52.92 ± 14.11	57.96 ± 13.20	0.003
Sex, no. (%)			0.697
Male	82 (68)	38 (32)	
Female	142 (72)	57 (28)	
Race, no. (%)			0.364
White	53 (29)	130 (71)	
Black	41 (32)	88 (68)	
Hispanic	1 (100)	0 (0)	
Asian	1 (100)	0 (0)	
Other	0 (0)	1 (100)	
BMI (kg/m^2^)	29.42 ± 8.10	30.14 ± 8.19	0.472
Obstruction (FEV_1_/FVC<70%), no. (%)	39 (40)	59 (60)	0.01
Restriction (TLC < 80%), no. (%)	33 (34.4)	63 (65.6)	0.238
Combined obstruction and restriction, no. (%)	3 (3.2)	2 (1.8)	0.444
FVC, L	2.88 ± 0.9	92.43 ± 1.05	0.001
FVC, %predicted	78.05 ± 16.51	68.55 ± 24.84	0.002
FEV_1_, L	2.10 ± 0.82	1.59 ± 0.74	<0.001
FEV_1_, %predicted	72.72 ± 21.55	57.64 ± 21.85	<0.001
FEV_1_/FVC ratio	73.14 ± 15.80	67.39 ± 21.38	0.027
TLC, L	4.94 ± 1.45	4.55 ± 1.55	0.05
TLC, % predicted	90.89 ± 20.44	85.08 ± 28.09	0.094
RV, L	2.10 ± 1.00	2.16 ± 1.10	0.636
RV, %predicted	114.85 ± 50.58	114.89 ± 58.69	0.996
DLCO, mL/min/mmHg	15.28 ± 5.92	9.77 ± 4.35	<0.001
DLCO, %predicted	68.54 ± 24.81	42.51 ± 19.25	<0.001
Airway resistance, cmH_2_O/L/sec	1.95 ± 1.33	2.20 ± 1.33	0.144

**Table 4 medsci-08-00008-t004:** The 6-minute walk test measurements in ≥4% oxygen Desaturators vs. Nondesaturators.

6MWT Variables	Nondesaturators (Δ SpO_2_ > −3%) *N* = 224 (70%)	Desaturators (Δ SpO_2_ ≤ −4%) *N* = 95 (30%)	*p*-Value
SpO_2_, %			
Start of test	97.56 ± 2.07	97.18 ± 2.43	0.157 ^1^
End of test	97.24 ± 2.26	89.67 ± 5.15	<0.001 ^1^
Change	−0.32	−7.51	<0.001 ^2^
Borg Dyspnea			
Start of test	1.26 ± 1.48	1.02 ± 1.29	0.163
End of test	3.26 ± 2.13	3.97 ± 7.55	0.196
Change	+2.00	+2.95	0.465
Borg Fatigue			
Start of test	1.46 ± 1.83	1.26 ± 1.57	0.354
End of test	3.52 ± 2.77	2.77 ± 2.28	0.012
Change	+2.06	+1.51	0.053
Pulse rate, 1/min			
Start of test	79.29 ± 15.30	80.93 ± 15.34	0.383
End of test	108.96 ± 57.87	106.33 ± 21.38	0.669
Change	+29.67	+25.40	0.848
6-minute walk distance, m	391.69 ± 128.83	352.67 ± 138.19	0.017 ^1^

^1^*p*-value from comparison based on categories of the change in SpO_2_ by ANOVA. ^2^
*p*-value from comparison based on categories of the change in SpO_2_ by Repeated Measures ANOVA.

**Table 5 medsci-08-00008-t005:** Linear regression model predicting the change in SpO_2_ from start to end of the 6-minute walk test.

Variable	Parameter Estimate (*β*) ± Standard Error	*p*-Value
(Constant)	4.886 ± 1.783	0.007
Age, years	0.002 ± 0.024	0.933
FVC, L	0.254 ± 0.79	0.748
FEV_1_, L	0.197 ± 1.022	0.847
DLCO, mL/min/mmHg	−0.291 ± 0.071	<0.001

**Table 6 medsci-08-00008-t006:** Logistic regression predicting an oxygen desaturation ≥ 3% during a 6-minute walk test.

Variable	Odds Ratio	95% Confidence Interval	*p*-Value
Age, years	0.951	0.892, 1.014	0.125
FVC, L	0.883	0.070, 11.203	0.923
FEV_1_, L	2.816	0.081, 98.292	0.568
DLCO, mL/min/mmHg	0.909	0.742, 1.115	0.360

**Table 7 medsci-08-00008-t007:** Logistic regression predicting an oxygen desaturation ≥ 4% during a 6-minute walk test.

Variable	Odds Ratio	95% Confidence Interval	*p*-Value
Age, years	1.003	0.973, 1.034	0.859
FVC, L	0.581	0.174, 1.933	0.376
FEV_1_, L	3.574	0.751, 17.014	0.110
DLCO, mL/min/mmHg	0.749	0.655, 0.856	<0.001

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
