# Peer review of "Exercise-Induced Oxygen Desaturation during the 6-Minute Walk Test"

_medsci, 2020, doi:10.3390/medsci8010008_

Round 1

Reviewer 1 Report

This is an interesting paper, with important limitations but strong messages. Indeed, oxygen desaturation of effort does carry important pathophysiological information in PH. The data are however difficult to interpret without knowing the underlying diagnoses and no subgroup analyses.

I have the following comments:

How many patients had a known PFO? Do the authors routinely perform bubble echo studies in all PH patients? The pathophysiological implications of desaturation in pts with and without a shunt is quite different Methods, patient selection criteria: were patients with congenital heart disease and PH excluded? Were patients with low sats at rest included? Was there a relation between oxygen desaturation and haemoglobin concentration? Why use a cutoff of >=3% desaturation for the definition, then switch this to >=4% for “desaturators”?? In PH, desaturation is either a reflection of shunting or low cardiac output. Was there a relation to CO in this population? Or RV function? What was the background diagnosis in this population? How many PH/PAH/PVOD/COPD etc Are numbers in the large subgroups big enough to allow separate analyses (as that the pathophysiology is very different)? What was the severity of the disease in this population? When presenting prevalence of desaturation, it would be good to know the denominator. Saying this was probably moderate-to-very severe cardiopulmonary disease because they had a 6MWT is probably not enough (Line 227). What is the aim of the ROC analysis with cutoffs and sensitivity/specif? To identify patients who will desaturate from their DLCO? Why not just exercise them? It is probably cheaper. Line 262: Patients desaturating due to R-L shunts may not benefit from LTOT. Is the cut off or 3 or 4% appropriate for pulse oximetry? Perhaps a bigger cut off is needed for a technique that may become somewhat inaccurate on exercise? Is there a description of the pulse oximetry technique and device used in the Methods, technical limitations? Should desaturators not exclude patients in whom sats stayed above 90%? Is a drop from 99 to 95% true desaturation?

Reviewer 2 Report

The authors report that reduced DLCO predicts a decrease in hemoglobin saturation during a 6 MWT among a retrospective cohort of patients.  The data and results are interesting.  There is some confusion regarding the interpretation and also the presentation of the data.  Please see comments:

Abstract

define PFT

Main study

Line 62. What phenomenom?

Line 76. Spelling

Line 76. What is the clinical importance of this study?  This should be briefly stated in the intro.

Line 81. Exclusion criteria?  Previous diagnosis of COPD, on oxyge, what about medications?

Line 81. All from the same clinic?

Line 82. did you account for resting oxygen saturation? 

Line 114. So these were patients with already diagnosed pulmonary disease?

Line 144. This is very confusing because, the table shows that 35% had a 3% decrease?

Line 156. It is difficult to determine what the p-values represent in the table.  An association between the SaO2 and pulmonary or other measures?  Or, is it within a certain group?  Seems that the indicators should be given.

Line 170. "Meanwhile" is not a very scientific term.

Line 174. Why not visually mark these differences in the table?

Table 2. Were baseline values different?

Line 215. How many patients were already diagnosed with pulmonary disease? It seems that this would be an exclusion criterium?

Line 215. Where is the 19% coming from?

Line 226. In a brief search, several papers were located that measured oxygen desaturation after 6 MWT

Paciocco et al 2001, Eur Resp; Jenkins and Cecins 2010, Respirology; Jenkins and Cecins 2011 Int Med J; Chetta et al 2001.

It appears that exercise induced desaturation occurs often

Lines 255 – 261. What is meant here?  That desaturation during 6 MWT leads to supplemental oxygen use?  Also, if reduced DLCO leads to 6 MWT then why is this study important?  Meaning, what is the clinical value of this study?  Reduced DLCO will predict desaturation during 6 MWT, which will improve treatment?  Please clarify.

Author Response

Thank you for reviewing our manuscript. Please review the answers to your questions:

The authors report that reduced DLCO predicts a decrease in hemoglobin saturation during a 6 MWT among a retrospective cohort of patients.  The data and results are interesting.  There is some confusion regarding the interpretation and also the presentation of the data.  Please see comments:

Abstract

Define PFT

 We spelled out and defined PFT as pulmonary function test (i.e., spirometry, lung volumes, diffusion capacity) in the abstract

Main study

Line 62. What phenomenon?

 Exercise-induced desaturation (EID)

Line 76. Spelling

 Spelling corrected.

Line 76. What is the clinical importance of this study?  This should be briefly stated in the intro.

 We added the following statement in the Introduction: “Knowledge of which PFT measurements predict O2 desaturation during exercise can assist the clinician in identifying patients who may require oxygen supplementation during exercise in order to relieve dyspnea and to enhance exercise capacity.”

Line 81. Exclusion criteria?  Previous diagnosis of COPD, on oxygen, what about medications?

We added the statement: “The only exclusion criterion was the absence of either PFT or 6MWT results.”

Line 81. All from the same clinic?

Yes, all patients were referred from the Saint Louis University Hospital or ambulatory clinic (SLUCare).

Line 82. did you account for resting oxygen saturation?

Yes, the change in SpO2 was calculated as the difference between the SpO2 at rest at the beginning of the 6MWT test and at then end of the test.  In Table 2, we listed SpO2 at the start of the 6MWT as an independent variable.

Line 114. So these were patients with already diagnosed pulmonary disease?

Some patients have previously diagnosed cardiopulmonary disease while other patients were undergoing their initial evaluation for suspected cardiopulmonary disease based on symptoms and clinical findings.

Line 144. This is very confusing because, the table shows that 35% had a 3% decrease?

We apologize for the confusion.  The prevalence of a 3% drop in SpO2 was 35%.  We have corrected the values in the Results and Discussion accordingly

Line 156. It is difficult to determine what the p-values represent in the table.  An association between the SaO2 and pulmonary or other measures?  Or, is it within a certain group?  Seems that the indicators should be given.

 The p values in Tables 1 and 2 indicate whether the independent variables (age, sex, race, BMI, PFT measurements, 6MWT measurements) were statistically significantly different among those whose SpO2 increased by at least 1%, decreased by 3%, or did not change (-2% to 0% difference from baseline).  These tables attempt to identify which independent variables may be associated with O2 desaturation using a univariate analysis.

Line 170. "Meanwhile" is not a very scientific term.

 We have removed the word, “Meanwhile,” for the manuscript.

Line 174. Why not visually mark these differences in the table?

Table 2. Were baseline values different?

Yes, the baseline SpO2 were statistically significantly different between the groups as listed on Table 2:

               Increased SpO2 group:  baseline SpO2 = 96.57%

             Unchanged SpO2 group: baseline SpO2 = 97.99%

             Decreased SpO2 group: baseline SpO2 = 97.20%

Although the values were statistically significantly different, there was no meaningful clinical pattern in the difference in values.

 Line 215. How many patients were already diagnosed with pulmonary disease? It seems that this would be an exclusion criterium?

Since this was a retrospective study and we did not clinically evaluate the patients ourselves, we did not ascertain the specific diagnoses.  In addition, some patients were undergoing PFT and 6MWT as part of an initial evaluation of their respiratory or cardiovascular symptoms and did not yet have an established diagnosis at the time of testing.

Line 215. Where is the 19% coming from?

We apologize: the 19% was a typographic error.  The correct prevalence of patients who experienced at least a 3% drop in oxygen saturation was 35% (113 of 319 participants) as listed on Tables 1 and 2.  We have corrected the manuscript accordingly.

Line 226. In a brief search, several papers were located that measured oxygen desaturation after 6 MWT

Paciocco et al 2001, Eur Resp; Jenkins and Cecins 2010, Respirology; Jenkins and Cecins 2011 Int Med J; Chetta et al 2001.

It appears that exercise induced desaturation occurs often

Thank you so much for sending us the references on 6MWT and oxygen desaturation.  We reviewed those references and have incorporated the findings of those previous studies in our introduction and discussion accordingly as follows:

Introduction:

“Previous studies have observed O2 desaturation to occur commonly during the 6MWT.   Paciocco et reported that 18 of 34 (53%) of pulmonary hypertensive patients experienced a ≥10% O2 desaturation during 6MWT and an almost 3-fold increased mortality over a 5-year follow-up.  Jenkins et al reported a 47% prevalence of significant O2 desaturation defined as a ≥4% fall in SpO2 to < 90% in a large cohort of patients with chronic lung disease, with pre-exercise SpO2 and forced expiratory volume in 1 second (FEV1) as significant predictors of desaturation.  An analysis of 6MWT in cystic fibrosis patients with mild-to-moderate lung disease by Chetta et al identified forced expiratory volume in 1 second (FEV1) as the only independent predictor of SpO2 during a 6MWT.    Knowledge of which PFT measurements predict O2 desaturation during exercise can assist the clinician in identifying patients who may require O2 supplementation during exercise in order to relieve exertional dyspnea and to enhance exercise capacity.”

Discussion:

“The 35% prevalence of ≥3% O2 desaturation corroborates the common occurrence of EID during a 6MWT in other studies:  53% prevalence of ≥10% desaturation in pulmonary hypertensive patients in the Paciocco study   and 47% prevalence of a 4% desaturation in patients with chronic lung disease in the Jenkins study .  The lower prevalence of O2 desaturation in our cohort with various cardiopulmonary disorders may be due to the inclusion of patients with less severe pulmonary function derangement in our cohort.”

Lines 255 – 261. What is meant here?  That desaturation during 6 MWT leads to supplemental oxygen use?  Also, if reduced DLCO leads to 6 MWT then why is this study important?  Meaning, what is the clinical value of this study?  Reduced DLCO will predict desaturation during 6 MWT, which will improve treatment?  Please clarify.

Yes, since we found that a reduced DLCO is a statistically significant predictor of O2 desaturation during 6MWT, a reduced DLCO may be a useful clinical indicator of the need for O2 during exertion.  Specifically, a DLCO of 45% may be a useful clinical cut-off for ordering 6MWT or oxygen desaturation study to determine whether patient has exercise-induced hypoxemia requiring O2 therapy.

Round 2

Reviewer 2 Report

It is difficult to identify where the revisions were made as the lines numbers were not included.  Also, the what does the red font indicate?  Editing of the entire manuscript.  Lastly, the authors did not revise the tables, and so it remains difficult to understand the findings.

Line 74, please define O2 desaturation.

Line 103. The connection between O2 desaturation during 6MWT and exercise tolerance is still unclear.  The intro has emphasized mortality predictors, but the purpose of monitoring O2 saturation as a method to improve exercise tolerance is still unclear.  Do patients who are monitored during a 6MWT do better or worse in cardiac/pulmonary rehab if they   desaturate during the 6MWT?  The purpose is to determine if PFT predict EID during 6MWT; however, maybe PFT measures in the absence of a 6MWT among cardiac/pulmonary rehab patients may assist clinicians in predicting EID. 

The authors made no revisions to the tables to improve readership.  Why report all three categories if the p-value is an association for the general category. 

Explaining the statistical values should be improved.  Not just explained to the reviewer, but revised to improve the general reader.  Not sure why revisions were not made.  Also, the baseline values differences should be reported.

Author Response

Please find below the responses to the reviewer’s

It is difficult to identify where the revisions were made as the lines numbers were not included.  Also, the what does the red font indicate?  Editing of the entire manuscript.  Lastly, the authors did not revise the tables, and so it remains difficult to understand the findings.

 We have included the line numbers for this revision to help reviewer identify where the revisions were made.

The red font tracks the changes made to the manuscript.

We have revised Tables 1, specifically the percentages of the categorical variables age, sex, no. and percentage of participants with obstruction, restriction, and combined obstruction-restriction, respectively.  The original percentages reported were the % within the desaturation category instead of within the age and sex, respectively.  For sex, we also added the n (%) for the females so that the values can be compared visually with the males.  Tables 1 & 2 enable the reader to compare the mean values (for continuous variables) and numbers and percentages (for categorical variables) among the nondesaturators, unchanged SpO2, and 3% desaturator groups.  The p values reflect which independent variable mean values statistically significantly vary among the 3 groups categorized based on change inSpO2.

Line 74, please define O2 desaturation.

The Andrianapoulos article (reference 5) defined exercise-induced oxygen desaturation as SpO2 ≤88%.  We have added this definition on line 74.

Line 103. The connection between O2 desaturation during 6MWT and exercise tolerance is still unclear.  The intro has emphasized mortality predictors, but the purpose of monitoring O2 saturation as a method to improve exercise tolerance is still unclear.  Do patients who are monitored during a 6MWT do better or worse in cardiac/pulmonary rehab if they   desaturate during the 6MWT?  The purpose is to determine if PFT predict EID during 6MWT; however, maybe PFT measures in the absence of a 6MWT among cardiac/pulmonary rehab patients may assist clinicians in predicting EID. 

 Based on Table 2, patients whose SpO2 dropped by at least 3% had statistically significantly reduced walking distance compared to those whose SpO2 increased or remain unchanged from the start to the end of the 6MWT.

We did not investigate whether patients who underwent 6MWT do better or worse in cardiopulmonary rehab if they desaturation during the 6MWT because it was not in the scope of our study.  We concur that this would be a very good clinical question to answer in a future study.

We have found in our findings that although PFT measures are significantly associated with EID, they are not 100% accurate or precise in predicting EID.  A 6MWT is more accurate and precise since it directly measures SpO2 during exercise.

The authors made no revisions to the tables to improve readership.  Why report all three categories if the p-value is an association for the general category. 

We have revised Tables 1 & 2 as per the Reviewer’s suggestions in order to improve readability.

Tables 1 & 2 include the 3 categories to determine which the independent variables (age, sex, BMI, PFT values, 6MWT values) are statistically different based on the category of change in desaturation saturation (increased, no change, or decreased), to determine whether there is a crude association between these independent variables and the change in saturation category using ANOVA and Chi-square.  The use of 3 categories of O2 desaturation also attempts to demonstrate potential dose response between change in SpO2 and the independent variables.

Table 1 was corrected as described in the previous response.

Table 2 was revised so that the reader can visually compare the change in 6MWT parameters from the start to the end of the test for the 3 categories of O2 desaturation.  The p values reported from Repeated Measures ANOVA.

To clarify our findings further, we added Tables 3 and 4 which compare whether the independent variables (demographics, PFT, and 6MWT) were statistically different between the 4% desaturators vs the nondesaturators.  We also conducted Repeated Measures ANOVA to statistically compare the changes in 6MWT parameters from the start to the end of the 6MWT between the ≥4% desaturators and nondesaturators.

Explaining the statistical values should be improved.  Not just explained to the reviewer, but revised to improve the general reader.  Not sure why revisions were not made.  Also, the baseline values differences should be reported.

We have revised the description of the Statistical Analysis under Methods, explaining the addition of Repeated Measures ANOVA in comparing the changes in 6MWT results from start to end of the test among the 3 categories of SpO2 change (increased, unchanged, and decreased). 

We have added to additional tables (3 & 4) which compare demographic, clinical, PFT, and 6MWT values between ≥4% desaturators and nondesaturators.

We have added 2 sections to explain the results of Tables 3 & 4 under Results.

The baseline differences were added on the revised Table 2 so the reader can compare the change in 6MWT values among the 3 categories of SpO2 change (increased, unchanged, decreased) and the resulting p values of the comparisons using of the Repeat Measures ANOVA.

Round 3

Reviewer 2 Report

No further comments.  Nice work.